# The burden of cardiovascular disease attributable to dietary risk factors in Australia between 1990 and 2019

Sebastian V. Moreno[1], Riaz Uddin[1], Sarah A. McNaughton[1], Katherine M. Livingstone[1], Elena S. George[1], George Siopis[1], Ralph Maddison[1], Rachel R. Huxley[2], Sheikh Mohammed Shariful Islam[1]*

**1** Institute for Physical Activity and Nutrition (IPAN), School of Exercise and Nutrition Sciences, Deakin University, Geelong, VIC, Australia, **2** Faculty of Health, Deakin University, Geelong, VIC, Australia

☉ These authors contributed equally to this work.
* shariful.islam@deakin.edu.au

**Data Availability Statement:** The datasets used in the current study are publicly available at https://vizhub.healthdata.org/gbd-results/. Data were extracted using the GBD Compare and GBD

## Abstract

Unhealthy diet is associated with increased risk of cardiovascular diseases (CVD). However, there are no studies reporting the impact and trends of dietary risk factors on CVD in Australia. This study aimed to determine the burden of CVDs attributable to dietary risk factors in Australia between 1990 and 2019. We used data from the Global Burden of Diseases (GBD) study and quantified the rate (per 100,000) of deaths, disability-adjusted life years (DALYs), years lived with a disability (YLDs), and years of life lost (YLLs) for 21 CVDs attributable to 13 dietary risk factors (eight food groups and five nutrients) in Australia by sex and age groups (≥25 years and over). In 2019, the age-standardised rates of deaths, YLDs, YLLs, and DALYs attributable to dietary risk factors attributable to CVDs in the Australian population were 26.5, 60.8, 349.9, and 410.8 per 100,000 in women and 46.1, 62.6, 807.0, and 869.6 in men. Between 1990 and 2019, YLLs consistently contributed more towards the rates of DALYs than YLDs. Over the 30-year period, CVD deaths, YLLs, and DALYs attributable to dietary risk factors declined in both women and men. The leading dietary risk factors for CVD deaths and DALYs were a diet high in red meat (6.1 deaths per 100,000 [3.6, 8.7] and 115.6 DALYs per 100,000 [79.7, 151.6]) in women and a diet low in wholegrains (11.3 deaths [4.4, 15.1] and 220.3 DALYs [86.4, 291.8]) in men. Sex differences were observed in the contribution of dietary risk factors to CVD over time such that the lowest rate of decrease in deaths and DALYs occurred with diets high in sodium in women and diets high in processed meat in men. Although the burden of diet-related CVD has decreased significantly in the Australian population over the past 30 years, diets low in wholegrains and high in red meat continue to contribute significantly to the overall CVD burden. Future nutrition programs and policies should target these dietary risk factors.

Results tool (https://www.healthdata.org/data-tools-practices/interactive-visuals/gbd-compare). Additional general data are available from the Global Burden of Disease: https://www.healthdata.org/gbd/2019.

**Funding:** The authors received no specific funding for this work.

**Competing interests:** The authors have declared that no competing interests exist.

## Introduction

Cardiovascular diseases (CVDs) continue to be the leading causes of disability and premature death in the Australian population [1, 2]. Over the last 20 years, ischemic heart disease consistently ranked among the leading causes for loss of quality of life and premature death in Australia. While the number of CVD-related deaths has decreased since the peak of the Australian CVD epidemic (from 33,411 deaths in 1968 to 18,590 in 2017), CVDs remain the major causes of poor health and rising healthcare expenditure. In 2017–18, the Australian Health Survey reported that 1.2 million adults had one or more heart or blood vessel-related conditions, with 1.2 million (11%) Australians hospitalised due to a cardiovascular condition such as ischemic heart disease, heart failure, cardiomyopathy, ischemic stroke.[1] CVDs were estimated to cost the Australian healthcare system $10.4 billion, the second-highest behind musculoskeletal disorders ($12.5 billion) [1, 3].

CVDs are attributed to a range of cardiometabolic, environmental, social, and behavioural risk factors. A suboptimal diet–defined as either overconsumption of an unhealthy dietary pattern or underconsumption of a healthy dietary pattern–is a strong predictor of poor cardiovascular health across the lifespan [4]. The relationship between CVDs and dietary patterns, food groups, and nutrients has been studied comprehensively and is well-established.[5] A recent study[5] reported that CVDs were the leading causes of diet-related death, disability-adjusted life years (DALYs) globally, ahead of cancer and type 2 diabetes (T2D). The current scientific consensus indicates that diets high in wholegrains, fruits, vegetables, nuts and seeds, legumes, unsaturated fatty acids (i.e. polyunsaturated fatty acids and monounsaturated fatty acids), dietary fibre, and diets low in processed meats, red meat, sugar-sweetened beverages (SSB), saturated fatty acids, trans-fatty acids, and moderate dietary sodium can reduce the risk of cardiovascular deterioration and complications [5–7]. Most Australians do not meet the minimum recommended serves for the five major foods groups (i.e. grains, vegetables/legumes/beans, lean meats, fruit, and dairy) and overconsume discretionary foods (those high in salt, fat and sugar) [8, 9]. While the Australian Dietary Guidelines (ADG) provide evidence-based recommendations to reduce chronic disease prevalence [10], recent findings indicate that many Australians continue to have a suboptimal dietary pattern that promotes CVDs and other non-communicable diseases (NCD) [11, 12].

Although several studies have investigated diet-related CVD burden, no previous study has investigated the trend in CVDs attributable to dietary risk factors in Australia [7, 13–15]. A systematic analysis of long-term trends in diet-related CVD burden is essential to provide a thorough understanding of the impact of individual dietary risk factors on the CVD burden in Australia and to inform future public health policy. Therefore, this study aims to examine the trends of CVD burden attributable to dietary risk factors between 1990 and 2019 in the Australian population and compare diet-related CVD burden between sex and age groups.

## Materials and methods

### Data sources

We used data from the Global Burden of Diseases, Injuries, and Risk Factors Study (GBD) and quantified the burden of 21 CVDs attributable to 13 dietary risk factors in adults (≥25 years) by sex and 15 age groups between 1990 and 2019 in Australia. GBD data are available for 23 age groups; however, as CVDs are rare in the younger population and are more likely caused by genetic conditions, we included all adults over the age of 25 years in successive 5-year age bands (e.g. 25–29, 30–34 years). Adults aged ≥95 years were grouped in one band.

## Study variables

In this study, we included 21 underlying causes of cardiovascular mortality and morbidity and 13 related dietary risk factors, which are available from the GBD 2019. Each CVDs or cardiovascular conditions were identified with standard case definitions [16]. Details of the 21 CVDs investigated in this study are summarised in S1 Table in S1 File). The GBD 2019 dietary risk factors comprise 15 food types, which are either under consumed (i.e. low intake of calcium, fibre, fruits, legumes, milk, nuts and seeds, omega-3 fatty acids from seafood, polyunsaturated fatty acids, vegetables, or wholegrains) or overconsumed (i.e. high intake of processed meat, red meat, sodium, sugar-sweetened beverages, or trans-fatty acids). Data for a diet low in calcium or milk were not available for Australia in the GBD Compare. Therefore, we included 13 dietary risk factors. Consumption data were collected by multiple sources, including nutrition surveys, household budget surveys, and United Nations Food and Agricultural Organization (FAO) Food Balance Sheets and Supply and Utilization Accounts (details of data sources available at GHDx website). For sodium and trans fatty acids, we used available data from 24-hour urinary sodium and partially hydrogenated vegetable oil in packaged foods. All dietary data (except for sodium and SSB) were standardised to 2000 calories/day by GBD. We presented the dietary risk factor definitions in S2 Table in S1 File.

## Data presentation

We estimated the standard epidemiological and summary measures of mortality and morbidity, which included the rate of deaths, disability-adjusted life years (DALYs), years lived with a disability (YLDs), and years of life lost (YLLs). GBD used vital registration data coded to the International Classification of Disease (ICD) system or verbal autopsy data to estimate the cause of death. YLLs are years lost due to premature mortality and are based on a reference maximum observed life expectancy [16]. YLDs are years lived in poor health or with a disability and are based on standardised disability weights for each health condition. DALYs are computed as the sum of YLLs and YLDs [16]. These health matrices are estimated using life tables, prevalence estimates, and disability weights. In this study, we produced age-group-specific and age-standardised estimates as a rate per 100,000 population. GBD used the direct method to perform age standardisation based on the global age structure from 2019, (see S1 Appendix in S1 File).

## Data analysis

The GBD 2019 used a six-step comparative risk assessment framework to estimate the attributable burden of risk factors [17]. At the first step, dietary risk-CVD outcome pairs (e.g. diet high in sodium and ischemic heart disease) were identified for inclusion. Next, for each risk-outcome pair, the relative risk was estimated as a function of exposure using systematic reviews and meta-regressions. The third step involved estimation of the distribution of exposure for each risk by age-sex-geography-time. Bayesian statistical models such as spatiotemporal Gaussian process regression, DisMod-MR 2.1 or network meta-regression were applied for the estimation. At the next step, the level of exposure with minimum risk called the theoretical minimum risk exposure level (TMREL) was determined for each dietary risk factor based on published trials or cohort studies. Each risk-outcome pair was weighted by the relative global magnitude of each outcome. The TMRELs for dietary risk factors included in this study are presented in S2 Table in S1 File. In the fifth step, the attributable deaths, YLLs, YLDs, and DALYs were estimated by the multiplying population attributable fraction (PAF) by the relevant outcome quantity across age, sex, time, and geography. Finally, the PAF and attributable burden for combinations of risk factors were estimated using a mediation matrix, which took

into account the mediation of different risk factors through other risk factors. The models were controlled and adjusted for bias in data and other types of information, such as country-level covariates. The final attributable burden estimates included uncertainty in each step of the analysis. Details of the modelling approaches used in this study can be found elsewhere [16, 17].

The GBD repeated all calculations 1,000 times using one draw of each parameter at each iteration when estimating the burden. Using these 1,000 draws, the median (point estimate) and 95% uncertainty intervals (2.5th and 97.5th percentiles of distribution) for the final estimates were calculated. Two estimates for a parameter (e.g., DALYs for males vs females or DALYs for the year of 2016 vs 2017) are considered statistically and significantly different if their 95% Uis do not overlap [5]. GBD received approval from the University of Washington Institutional Review Board Committee (STUDY00009060).

## Results

### Diet-related CVD mortality and morbidity

In 1990, a total of 20,706 (95% UI 17399, 23842) CVD deaths, 18189 (12457, 25240) YLDs, 377030 (320870, 430311) YLLs, and 395219 (335114, 451443) DALYs were attributable to dietary risk factors in Australia. The respective numbers of CVD deaths, YLDs, YLLs, and DALYs were 16667 (13496, 19837), 23739 (16246, 33228), 233290 (194121, 272314), and 257030 (215440, 303699) in 2019. In 2019, the overall age-standardised rates of deaths, YLDs, YLLs, and DALYs attributable to dietary risk factors were 35.7 (95% UI 29.2, 42.2), 61.5 (42.0, 86.5), 569.3 (477.9, 662.9), and 630.8 (532.9, 743.6) per 100,000 population, respectively (Fig 1).

### Trends in diet-related CVD burden

In Australia, between 1990 and 2019, the age-standardised rates of CVD deaths, YLLs, and DALYs attributable to dietary risk factors were significantly higher in men than women (Fig 1). In both sexes, the rate of deaths, YLDs, YLLs, and DALYs decreased significantly between 1990 and 2019 (Table 1). Over this period, the rate of all CVD deaths attributable to dietary risk factors decreased by 68.3% (from 146 to 46 per 100,000) in men and 68.4% (from 83 to 27 deaths in 2019) in women. The rate of all CVD YLDs decreased by 37.1% (from 99.5 to 62.6) in men and 33.1% (from 90.9 to 60.8 per 100,000) in women, the rate of YLLs decreased by 72.5% (from 1,272.6 to 349.9 per 100,000) in women and 70.5% (from 2,738.7 to 807.0) in men, and rate of DALYs decreased by 69.9% (from 1,363.5 to 410.8 per 100,000) in women and 69.4% (from 2,838.3 to 869.6) in men (Table 1 and S3 Table in S1 File).

### Trends in diet-related CVD risk factors

Between 1990 and 2019, the leading dietary risk factors for CVD deaths, YLLs, and DALYs per 100,000 were a diet high in red meat (6.1 deaths [3.6, 8.7], 90.6 YLLs [57.3, 122.9], and 115.6 DALYs [79.7, 151.6]) in women and a diet low in wholegrains (11.3 deaths [4.4, 15.1], 207.6 YLLs [79.9, 275.1], and 220.3 DALYs [86.4, 291.8]) in men (Table 1). Between 1990 and 2019, a diet high in red meat was the leading dietary risk factor for CVD YLDs for both women and men. Of the 13 dietary risk factors assessed, diets high in sodium for women and diets high in processed meat for men observed the lowest decrease in percentage change for age-standardised rate for CVD deaths, YLLs, and DALYs between 1990 and 2019 (S3 Table in S1 File). Diets high in sodium had the lowest decrease in the rate of YLDs per 100,000 in both sexes during this time: 18.3% in women (from 56.0 to 19.4) and 24.8% reduction in men (from 188.9

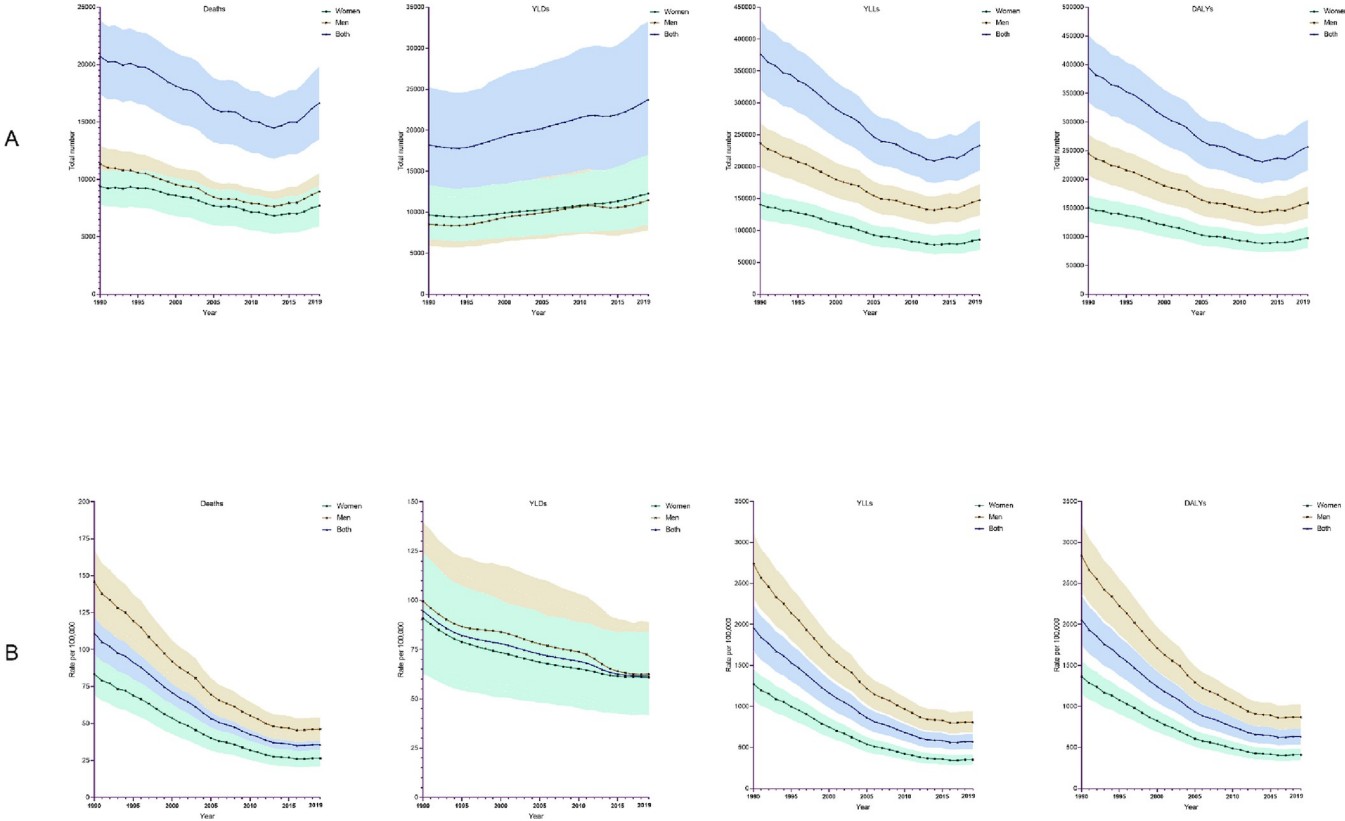

**Fig 1.** (**A**) The numbers of cardiovascular disease deaths, years lived with disability (YLDs), years of life lost (YLLs), and disability-adjusted life years (DALYs), and (**B**) the age-standardised rate (per 100,000) of deaths, YLDs, YLLs, and DALYs attributable to dietary risk factors in Australia between 1990 and 2019, overall and by sex. **Note** the shaded area represents 95% uncertainty intervals.

to 64.3). In both women and men, the highest decrease in the rate of YLDs, YLLs, and DALYs associated with dietary risk factors was for a diet low in nuts and seeds (S3 Table in S1 File).

## Major CVDs attributable to dietary risk factors

For all CVDs, ischemic heart disease had the highest rates of deaths, YLLs, and DALYs attributable to dietary risk factors in both sexes in 2019 (Table 2). The rates of death, YLLs, and DALYs due to ischemic heart disease attributable to dietary risk factors decreased significantly in both sexes between 1990 and 2019 (Table 2 and S4 Table in S1 File): the rate of deaths per 100,000 decreased by 70.2% (from 69 to 21) in women and 69.3% (from 69 to 40) in men, the rate of YLLs decreased by 74.7% (from 1043.8 to 264.6) in women and 71.3% (from 2432.2 to 697.2) in men, and the rate of DALYs decreased by 73.7% (from 1081.1 to 283.9) in women and 70.7% (from 2488.2 to 730.1) in men.

## Age-specific burden of diet-related CVDs

The burden of CVD deaths, YLDs, YLLs, and DALYs attributable to dietary risk factors was higher in the older age groups in both sexes (Table 3). The rates in each age group, however, were higher in men than women. In 2019, the highest rates of deaths, YLDs, YLLs, and DALYs were in women and men aged ≥95 years. Between 1990 and 2019, the rates of deaths, YLLs, and DALYs decreased significantly in all age groups in both sexes; however, the change in rates for YLDs was non-significant in women aged 25–29, 35–39, 40–44, 45–49, and 50–54

**Table 1. Burden of cardiovascular disease deaths, years lived with disability (YLDs), years of life lost (YLLs), and disability-adjusted life years (DALYs) attributable to overall and specific dietary risk factors in Australia, by sex, in1990 and 2019.**

| Dietary risk factor | Sex | Age-standardised rate of deaths (95% UI) per 100,000 | | Age-standardised rate of YLDs (95% UI) per 100,000 | | Age-standardised rate of YLLs (95% UI) per 100,000 | | Age-standardised rate of DALYs (95% UI) per 100,000 | |
|---|---|---|---|---|---|---|---|---|---|
| | | **1990** | **2019** | **1990** | **2019** | **1990** | **2019** | **1990** | **2019** |
| All risk factors combined | Men | 146.0 | 46.1 | 99.5 | 62.6 | 2738.7 | 807.0 | 2,838.3 | 869.6 |
| | | (122.6, 167.6) | (38.1, 54.2) | (68.8, 139.5) | (42.6, 88.9) | (2316.6, 3110.6) | (678.5, 942.0) | (2399.1, 3235.5) | (727.9, 1023.8) |
| | Women | 83.3 | 26.5 | 90.9 | 60.8 | 1272.6 | 349.9 | 1363.5 | 410.8 |
| | | (69.1, 96.8) | (20.8, 32.0) | (62.6, 123.7) | (41.6, 84.1) | (1071.6, 1458.9) | (287.4, 413.2) | (1149.5, 1561.3) | (342.1, 487.0) |
| Diet low in fruit | Men | 14.7 | 4.5 | 12.6 | 7.7 | 285.3 | 82.0 | 297.9 | 89.7 |
| | | (6.9, 21.0) | (2.1, 6.5) | (6.3, 20.4) | (3.8, 12.7) | (129.0, 405.0) | (38.2, 118.9) | (138.6, 422.3) | (43.8, 128.8) |
| | Women | 9.1 | 2.8 | 14.2 | 9.8 | 147.4 | 41.0 | 161.6 | 50.8 |
| | | (4.7, 12.8) | (1.4, 4.1) | (7.1, 23.4) | (4.6, 16.5) | (77.9, 206.2) | (22.1, 58.3) | (87.4, 224.8) | (28.6, 72.8) |
| Diet low in vegetables | Men | 13.5 | 3.9 | 9.0 | 4.9 | 256.5 | 67.1 | 265.5 | 72.0 |
| | | (6.7, 20.2) | (1.9, 5.7) | (4.5, 14.1) | (2.3, 7.9) | (124.4, 385.0) | (30.1, 99.9) | (129.6, 396.0) | (33.3, 106.1) |
| | Women | 7.6 | 2.2 | 8.4 | 4.8 | 117.5 | 28.8 | 125.8 | 33.6 |
| | | (3.9, 11.1) | (1.1, 3.2) | (3.9, 13.3) | (2.0, 8.1) | (59.8, 171.1) | (13.7, 42.8) | (65.2, 180.8) | (16.7, 50.1) |
| Diet low in legumes | Men | 33.1 | 10.1 | 14.6 | 8.4 | 640.6 | 181.3 | 655.2 | 189.7 |
| | | (8.7, 54.4) | (2.5, 16.8) | (3.4, 26.8) | (1.8, 15.6) | (163.2, 1048.0) | (41.1, 303.3) | (167.0, 1072.0) | (43.1, 318.7) |
| | Women | 17.4 | 5.2 | 9.5 | 4.8 | 265.9 | 66.6 | 275.4 | 71.4 |
| | | (4.2, 29.0) | (1.1, 8.7) | (2.1, 17.8) | (1.0, 9.0) | (63.7, 446.7) | (14.3, 112.3) | (65.7, 462.2) | (15.4, 120.6) |
| Diet low in wholegrains | Men | 34.3 | 11.3 | 19.5 | 12.7 | 662.7 | 207.6 | 682.3 | 220.3 |
| | | (13.7, 45.8) | (4.4, 15.1) | (9.2, 29.2) | (6.1, 18.7) | (261.5, 880.9) | (79.9, 275.1) | (270.9, 903.9) | (86.4, 291.8) |
| | Women | 17.9 | 5.9 | 15.6 | 10.7 | 272.9 | 77.1 | 288.5 | 87.8 |
| | | (7.5, 24.5) | (2.5, 8.2) | (7.7, 23.6) | (5.2, 16.4) | (112.6, 369.1) | (31.6, 104.4) | (123.1, 391.1) | (38.5, 118.8) |
| Diet low in nuts and seeds | Men | 8.6 | 2.2 | 3.7 | 1.6 | 154.9 | 33.7 | 158.6 | 35.2 |
| | | (2.7, 14.3) | (0.9, 3.8) | (1.1, 6.8) | (0.5, 3.0) | (48.3, 259.3) | (12.2, 58.3) | (49.2, 265.4) | (12.6, 61.3) |
| | Women | 3.8 | 1.1 | 1.9 | 0.7 | 53.2 | 11.1 | 55.2 | 11.8 |
| | | (1.1, 7.0) | (0.4, 2.0) | (0.5, 3.8) | (0.2, 1.5) | (15.0, 96.7) | (3.7, 20.2) | (15.4, 99.7) | (3.9, 21.6) |
| Diet high in red meat | Men | 30.6 | 10.0 | 30.1 | 19.7 | 620.8 | 191.0 | 650.8 | 210.7 |
| | | (15.6, 44.8) | (5.4, 14.5) | (18.8, 42.0) | (12.5, 27.6) | (322.8, 895.6) | (107.3, 272.2) | (343.9, 928.6) | (123.5, 294.1) |
| | Women | 18.1 | 6.1 | 33.5 | 26.0 | 302.8 | 90.6 | 336.3 | 115.6 |
| | | (10.2, 25.9) | (3.6, 8.7) | (22.4, 45.8) | (16.9, 34.2) | (179.6, 416.5) | (57.3, 122.9) | (209.3, 433.7) | (79.7, 151.6) |
| Diet high in processed meat | Men | 8.3 | 3.1 | 3.8 | 2.9 | 180.7 | 66.7 | 184.5 | 69.6 |
| | | (0.6, 18.5) | (0.3, 6.3) | (0.3, 8.9) | (0.2, 6.1) | (12.3, 383.5) | (5.4, 127.4) | (12.6, 393.3) | (5.7, 132.8) |
| | Women | 5.1 | 1.7 | 3.0 | 1.9 | 85.3 | 25.6 | 88.3 | 27.5 |
| | | (0.4, 10.6) | (0.2, 3.5) | (0.3, 6.4) | (0.2, 3.8) | (7.2, 170.4) | (2.6, 49.0) | (7.4, 176.7) | (2.8, 52.8) |
| Diet high in SSB | Men | 4.4 | 1.5 | 2.0 | 1.3 | 89.7 | 29.4 | 91.7 | 30.7 |
| | | (1.1, 7.2) | (0.4, 2.4) | (0.5, 3.8) | (0.3, 2.4) | (20.2, 144.6) | (6.3, 47.5) | (20.6, 147.6) | (6.7, 50.0) |
| | Women | 2.6 | 0.8 | 1.5 | 0.8 | 40.8 | 11.6 | 42.3 | 12.5 |
| | | (0.6, 4.5) | (0.2, 1.5) | (0.3, 2.8) | (0.2, 1.6) | (9.0, 69.6) | (2.4, 19.6) | (9.3, 72.0) | (2.5, 20.9) |
| Diet low in fibre | Men | 17.2 | 3.7 | 13.4 | 6.7 | 335.8 | 75.2 | 349.2 | 81.8 |
| | | (8.5, 26.4) | (1.7, 5.9) | (6.4, 21.6) | (3.0, 11.0) | (154.3, 514.2) | (34.7, 119.1) | (164.1, 531.6) | (38.3, 128.4) |
| | Women | 10.2 | 2.7 | 14.1 | 8.5 | 159.9 | 37.9 | 174.0 | 46.3 |
| | | (4.9, 15.6) | (1.3, 4.1) | (5.9, 23.3) | (3.1, 14.7) | (78.0, 243.7) | (18.8, 57.2) | (87.9, 263.4) | (24.0, 69.7) |
| Diet low in omega 3 fatty acids | Men | 7.4 | 2.3 | 3.2 | 1.8 | 138.5 | 39.1 | 141.7 | 40.9 |
| | | (5.1, 9.1) | (1.5, 2.9) | (1.9, 4.8) | (1.0, 2.7) | (95.5, 170.9) | (26.6, 49.5) | (98.0, 174.8) | (28.0, 52.2) |
| | Women | 4.2 | 1.2 | 2.3 | 1.1 | 62.6 | 15.6 | 64.9 | 16.6 |
| | | (2.5, 5.3) | (0.8, 1.6) | (1.3, 3.3) | (0.6, 1.7) | (40.6, 78.1) | (10.5, 19.7) | (42.3, 80.7) | (11.4, 21.2) |

(*Continued*)

**Table 1.** (Continued)

| Dietary risk factor | Sex | Age-standardised rate of deaths (95% UI) per 100,000 | | Age-standardised rate of YLDs (95% UI) per 100,000 | | Age-standardised rate of YLLs (95% UI) per 100,000 | | Age-standardised rate of DALYs (95% UI) per 100,000 | |
|---|---|---|---|---|---|---|---|---|---|
| | | **1990** | **2019** | **1990** | **2019** | **1990** | **2019** | **1990** | **2019** |
| Diet low in PUFA | Men | 9.6 | 2.9 | 4.3 | 2.5 | 187.6 | 53.2 | 191.8 | 55.6 |
| | | (1.0, 20.2) | (0.3, 6.1) | (0.4, 9.5) | (0.2, 5.6) | (18.1, 392.7) | (5.1, 111.6) | (18.6, 401.1) | (5.3, 116.7) |
| | Women | 5.3 | 1.5 | 2.9 | 1.5 | 81.5 | 20.3 | 84.4 | 21.8 |
| | | (0.6, 10.9) | (0.2, 3.2) | (0.3, 6.8) | (0.2, 3.4) | (8.4, 168.8) | (2.0, 42.5) | (8.8, 175.1) | (2.2, 45.2) |
| Diet high in trans fatty acids | Men | 124.1 | 6.7 | 10.6 | 5.6 | 464.6 | 121.5 | 475.1 | 127.2 |
| | | (1.6, 32.4) | (0.4, 9.1) | (0.6, 17.0) | (0.3, 9.1) | (29.4, 624.2) | (7.3, 163.6) | (29.8, 639.6) | (7.5, 171.1) |
| | Women | 12.9 | 3.5 | 7.0 | 3.3 | 197.2 | 45.7 | 204.2 | 49.0 |
| | | (1.0, 17.6) | (0.2, 4.8) | (0.5, 11.8) | (0.2, 5.5) | (15.3, 267.5) | (3.1, 62.1) | (15.7, 276.9) | (3.4, 66.3) |
| Diet high in sodium | Men | 8.8 | 3.1 | 11.5 | 8.7 | 188.9 | 64.3 | 200.4 | 72.9 |
| | | (0.8, 34.3) | (0.3, 12.1) | (0.8, 40.5) | (0.6, 30.5) | (13.5, 665.8) | (4.3, 218.0) | (14.4, 709.7) | (5.0, 246.6) |
| | Women | 3.5 | 1.4 | 6.8 | 5.6 | 56.0 | 19.4 | 62.8 | 24.9 |
| | | (0.5, 16.2) | (0.2, 6.2) | (0.7, 29.2) | (0.5, 22.6) | (6.9, 251.7) | (2.2, 84.0) | (7.8, 280.5) | (2.9, 106.0) |

*Abbreviation* UI = uncertainty interval

**Table 2. Burden of major cardiovascular disease deaths, years lived with disability (YLDs), years of life lost (YLLs), and disability-adjusted life years (DALYs) attributable to dietary risk factors in Australia, by sex, in 1990 and 2019.**

| Cardiovascular disease | Sex | Age-standardised rate of deaths (95% UI) per 100,000 | | Age-standardised rate of YLDs (95% UI) per 100,000 | | Age-standardised rate of YLLs (95% UI) per 100,000 | | Age-standardised rate of DALYs (95% UI) per 100,000 | |
|---|---|---|---|---|---|---|---|---|---|
| | | **1990** | **2019** | **1990** | **2019** | **1990** | **2019** | **1990** | **2019** |
| All cardiovascular diseases | Men | 146.0 | 46.1 | 99.5 | 62.6 | 2738.7 | 807.0 | 2838.3 | 869.6 |
| | | (122.6, 167.6) | (38.1, 54.1) | (68.8, 139.5) | (42.7, 89.0) | (2316.6, 3110.6) | (678.4, 942.0) | (2399.0, 3235.5) | (727.9, 1023.8) |
| | Women | 83.4 | 26.5 | 90.9 | 60.8 | 1272.6 | 349.9 | 1363.5 | 410.5 |
| | | (69.1, 96.8) | (20.8, 32.0) | (62.6, 123.7) | (41.6, 84.1) | (1071.6, 1458.9) | (287.7, 413.2) | (1149.5, 1561.2) | (342.1, 487.0) |
| Ischemic heart disease | Men | 128.5 | 39.5 | 56.0 | 32.9 | 2432.2 | 697.2 | 2488.2 | 730.1 |
| | | (106.2, 148.5) | (32.3, 46.3) | (35.5, 80.9) | (21.1, 48.0) | (2029.7, 2779.4) | (573.7, 801.7) | (2058.3, 2,844.7) | (602.6, 841.0) |
| | Women | 69.2 | 20.6 | 37.3 | 19.3 | 1043.8 | 264.6 | 1081.1 | 283.9 |
| | | (55.6, 80.9) | (16.0, 25.0) | (23.7, 55.0) | (12.2, 28.6) | (856.5, 1206.9) | (211.5, 311.7) | (884.8, 1,251.4) | (228.2, 335.1) |
| Stroke | Men | 16.7 | 6.2 | 39.3 | 25.7 | 290.8 | 100.3 | 330.2 | 126.0 |
| | | (13.3, 21.0) | (4.8, 8.0) | (26.2, 54.5) | (17.1, 35.7) | (233.5, 357.9) | (78.0, 126.0) | (265.8, 406.0) | (99.8,157.7) |
| | Women | 13.8 | 5.6 | 51.2 | 39.3 | 222.9 | 81.4 | 274.1 | 120.7 |
| | | (11.0, 16.9) | (4.2, 7.0) | (35.0, 69.6) | (26.2, 53.3) | (181.7, 268.8) | (63.9, 99.9) | (222.6, 332.6) | (95.3, 149.3) |
| Ischemic stroke | Men | 10.8 | 3.3 | 31.5 | 19.7 | 150.4 | 39.8 | 181.9 | 59.5 |
| | | (8.0, 14.0) | (2.3, 4.5) | (20.5, 44.4) | (13.0, 27.9) | (112.7, 191.7) | (28.6, 51.8) | (137.2, 230.8) | (44.0, 76.9) |
| | Women | 8.8 | 3.3 | 39.6 | 30.3 | 107.6 | 34.3 | 147.6 | 64.6 |
| | | (6.5, 11.2) | (2.3, 4.3) | (26.6, 55.3) | (19.6, 41.7) | (80.7, 134.9) | (24.6, 44.4) | (110.4, 185.8) | (46.5, 82.4) |
| Intracerebral haemorrhage | Men | 4.7 | 2.1 | 4.3 | 3.0 | 100.4 | 39.8 | 104.7 | 42.7 |
| | | (3.5, 6.1) | (1.5, 2.8) | (2.8, 6.2) | (1.9, 4.2) | (75.4, 128.0) | (29.1, 52.5) | (79.1, 133.4) | (31.4, 53.0) |
| | Women | 3.4 | 1.5 | 5.5 | 4.2 | 67.3 | 25.0 | 72.8 | 29.2 |
| | | (2.6, 4.4) | (1.0, 1.9) | (3.6, 7.8) | (2.7, 6.0) | (51.1, 85.0) | (18.4, 32.3) | (55.1, 92.3) | (21.6, 37.6) |

*Abbreviation* UI = uncertainty interval

**Table 3. Burden of cardiovascular disease deaths, years lived with disability (YLDs), years of life lost (YLLs), and disability-adjusted life years (DALYs) attributable to dietary risk factors in Australia, by age groups and sex between 1990 and 2019.**

| Age | Sex | Age-standardised rate of deaths (95% UI) per 100,000 | | Age-standardised rate of YLDs (95% UI) per 100,000 | | Age-standardised rate of YLLs (95% UI) per 100,000 | | Age-standardised rate of DALYs (95% UI) per 100,000 | |
|---|---|---|---|---|---|---|---|---|---|
| | | 1990 | 2019 | 1990 | 2019 | 1990 | 2019 | 1990 | 2019 |
| 25–29 years | Men | 3.1 | 1.4 | 21.3 | 18.5 | 188.8 | 83.0 | 210.0 | 101.5 |
| | | (2.5, 3.7) | (1.0, 1.8) | (13.4, 30.5) | (11.8, 26.9) | (155.8, 226.7) | (61.1, 109.7) | (174.03, 249.7) | (78.1, 130.5) |
| | Women | 1.4 | 0.4 | 35.8 | 33.0 | 86.7 | 27.2 | 122.5 | 60.2 |
| | | (1.1, 1.7) | (0.3, 0.6) | (21.4, 53.5) | (19.4, 49.4) | (70.3, 105.3) | (19.9, 36.1) | (98.6, 149.0) | (44.0, 80.7) |
| 30–34 years | Men | 6.7 | 3.4 | 29.7 | 25.0 | 381.7 | 194.8 | 411.4 | 219.8 |
| | | (5.7, 7.8) | (2.7, 4.3) | (17.5, 46.4) | (14.8, 39.6) | (320.3, 440.4) | (151.3, 242.7) | (345.9, 473.7) | (173.3, 270.9) |
| | Women | 2.6 | 1.1 | 47.0 | 42.7 | 144.5 | 63.1 | 191.5 | 105.8 |
| | | (2.1, 3.0) | (0.9, 1.4) | (29.2, 67.7) | (26.3, 62.0) | (119.1, 169.0) | (48.7, 79.8) | (155.3, 227.4) | (82.0, 132.5) |
| 35–39 years | Men | 16.9 | 8.0 | 43.6 | 35.5 | 873.3 | 413.5 | 916.9 | 449.0 |
| | | (14.7, 19.1) | (6.5, 9.7) | (27.4, 63.9) | (22.0, 53.4) | (757.6, 987.1) | (333.2, 500.6) | (801.4, 1037.4) | (367.8, 540.6) |
| | Women | 4.6 | 2.3 | 61.2 | 54.6 | 237.3 | 121.0 | 298.5 | 175.5 |
| | | (3.9, 5.4) | (1.9, 2.9) | (38.4, 87.3) | (33.0, 80.1) | (202.5, 276.3) | (97.1, 147.1) | (255.7, 351.2) | (140.2, 214.6) |
| 40–44 years | Men | 33.0 | 14.5 | 64.9 | 48.4 | 1543.2 | 676.4 | 1608.1 | 724.8 |
| | | (28.6, 37.1) | (11.7, 17.4) | (40.6, 95.5) | (28.5, 74.5) | (1338.5, 1735.8) | (546.0, 812.7) | (1385.4, 1809.7) | (590.6, 865.1) |
| | Women | 8.2 | 4.2 | 75.2 | 60.5 | 381.8 | 197.3 | 457.0 | 257.7 |
| | | (7.0, 9.5) | (3.4, 5.2) | (47.2, 110.9) | (37.6, 90.7) | (326.3, 442.6) | (160.3, 242.9) | (388.2, 533.4) | (207.8, 320.4) |
| 45–49 years | Men | 66.2 | 25.4 | 98.7 | 71.6 | 2767.9 | 1062.5 | 2866.6 | 1134.1 |
| | | (57.4, 73.9) | (20.5, 30.1) | (62.7, 144.4) | (45.7, 107.5) | (2402.3, 3091.9) | (860.3, 1259.9) | (2493.6, 3202.2) | (924.3, 1347.8) |
| | Women | 16.6 | 7.0 | 94.2 | 77.6 | 694.8 | 291.1 | 789.0 | 368.8 |
| | | (14.2, 19.0) | (5.6, 8.6) | (64.1, 134.0) | (49.3, 112.7) | (592.4, 793.8) | (233.1, 361.6) | (674.2, 905.3) | (298.0, 455.8) |
| 50–54 years | Men | 119.7 | 40.1 | 141.2 | 97.0 | 4439.2 | 1488.6 | 4580.3 | 1585.6 |
| | | (102.8, 136.2) | (33.0, 48.2) | (86.7, 206.2) | (59.9, 144.0) | (3811.4, 5047.8) | (1225.3, 1788.0) | (3929.1, 5209.0) | (1308.1, 1896.7) |
| | Women | 33.1 | 10.2 | 121.8 | 95.1 | 1229.3 | 379.8 | 1351.2 | 474.9 |
| | | (28.4, 38.0) | (8.1, 12.5) | (80.2, 174.2) | (59.8, 138.3) | (1052.1, 1408.5) | (302.1, 464.8) | (1161.2, 1550.2) | (386.7, 582.7) |
| 55–59 years | Men | 201.2 | 54.7 | 184.1 | 121.5 | 6517.6 | 1774.9 | 6701.7 | 1896.4 |
| | | (168.9, 229.2) | (44.8, 65.4) | (116.9, 274.9) | (73.2, 183.7) | (5471.8, 7425.5) | (1454.0, 2121.2) | (5615.9, 7661.8) | (1549.6, 2265.8) |
| | Women | 60.1 | 14.7 | 149.6 | 112.9 | 1946.5 | 477.7 | 2096.1 | 590.6 |
| | | (50.4, 69.1) | (11.9, 18.1) | (100.4, 214.2) | (73.1, 166.9) | (1632.1, 2239.9) | (385.8, 586.1) | (1759.1, 2422.2) | (479.8, 722.8) |
| 60–64 years | Men | 326.9 | 76.6 | 260.0 | 154.9 | 9092.1 | 2129.6 | 9352.1 | 2284.4 |
| | | (271.5, 376.5) | (62.2, 92.9) | (166.2, 377.2) | (95.6, 241.5) | (7550.7, 10472.3) | (1730.3, 2583.3) | (7770.7, 10772.1) | (1841.1, 2771.7) |
| | Women | 116.6 | 22.0 | 205.5 | 130.8 | 3240.6 | 612.1 | 3446.2 | 742.9 |
| | | (98.4, 133.5) | (17.7, 26.9) | (138.2, 284.7) | (84.8, 190.6) | (2736.8, 3711.4) | (492.4, 748.6) | (2921.9, 3934.0) | (602.6, 910.3) |
| 65–69 years | Men | 524.3 | 112.2 | 416.7 | 244.5 | 12253.7 | 2618.7 | 12670.4 | 2863.2 |
| | | (436.2, 608.9) | (91.2, 134.8) | (269.5, 609.7) | (154.2, 384.6) | (10194.6, 14229.7) | (2128.0, 3145.1) | (10518.2, 14787.0) | (2332.9, 3469.9) |
| | Women | 219.6 | 37.4 | 330.2 | 199.5 | 5121.6 | 872.4 | 5451.9 | 1071.9 |
| | | (183.8, 254.2) | (29.9, 45.8) | (221.3, 479.8) | (133.6, 288.3) | (4286.4, 5930.6) | (697.1, 1068.1) | (4571.9, 6323.8) | (866.1, 1320.6) |
| 70–74 years | Men | 809.2 | 176.8 | 581.2 | 340.1 | 15459.1 | 3368.1 | 16040.3 | 3708.2 |
| | | (668.2, 944.1) | (142.5, 212.9) | (372.2, 862.4) | (211.0, 537.3) | (12765.1, 18035.3) | (2715.0, 4055.7) | (13202.4, 18757.8) | (2972.4, 4477.1) |
| | Women | 416.9 | 72.7 | 477.7 | 275.8 | 7942.9 | 1383.4 | 8420.6 | 1659.3 |
| | | (344.7, 480.9) | (57.7, 89.6) | (314.4, 702.0) | (181.1, 408.1) | (6567.1, 9162.8) | (1097.2, 1705.3) | (7019.5, 9739.5) | (1322.4, 2039.8) |
| 75–79 years | Men | 1285.2 | 303.0 | 735.3 | 422.8 | 19447.6 | 4559.8 | 20182.9 | 4982.7 |
| | | (1061.8, 1519.6) | (243.3, 368.9) | (477.1, 1086.3) | (267.5, 656.0) | (16066.9, 22995.5) | (3661.3, 5550.9) | (16708.1, 23847.3) | (3982.7, 6136.0) |
| | Women | 755.7 | 158.0 | 632.5 | 349.0 | 11382.2 | 2371.5 | 12014.8 | 2720.4 |
| | | (619.8, 882.1) | (124.0, 196.3) | (417.5, 918.7) | (223.5, 514.2) | (9336.2, 13287.1) | (1861.4, 2946.0) | (9864.7, 14032.0) | (2154.1, 3377.0) |

*(Continued)*

**Table 3.** (Continued)

| Age | Sex | Age-standardised rate of deaths (95% UI) per 100,000 | | Age-standardised rate of YLDs (95% UI) per 100,000 | | Age-standardised rate of YLLs (95% UI) per 100,000 | | Age-standardised rate of DALYs (95% UI) per 100,000 | |
|---|---|---|---|---|---|---|---|---|---|
| | | **1990** | **2019** | **1990** | **2019** | **1990** | **2019** | **1990** | **2019** |
| 80–84 years | Men | 2004.6 | 599.9 | 917.4 | 508.0 | 23347.1 | 6920.3 | 24264.5 | 7428.3 |
| | | (1664.4, 2352.8) | (474.0, 727.8) | (596.8, 1332.1) | (313.9, 758.3) | (19385.6, 27402.6) | (5468.4, 8396.5) | (20026.5, 28558.5) | (5902.1, 9086.4) |
| | Women | 1382.1 | 381.1 | 837.0 | 459.7 | 15989.4 | 4378.8 | 16826.4 | 4838.5 |
| | | (1128.3, 1621.0) | (287.8, 473.3) | (559.1, 1181.1) | (301.2, 661.4) | (13053.7, 18752.9) | (3306.8, 5438.5) | (13752.2, 19658.0) | (3727.2, 6018.8) |
| 85–89 years | Men | 2799.1 | 1236.3 | 1086.3 | 641.4 | 24856.0 | 10890.4 | 25942.3 | 11531.8 |
| | | (2237.2, 3347.6) | (961.5, 1524.6) | (670.1, 1607.8) | (389.5, 979.9) | (19865.9, 29726.6) | (8469.4, 13429.7) | (20782.9, 31078.6) | (8968.5, 14291.1) |
| | Women | 2276.2 | 902.2 | 1025.8 | 575.5 | 20072.0 | 7904.5 | 21097.7 | 8480.0 |
| | | (1748.0, 2754.1) | (654.2, 1125.7) | (669.5, 1493.2) | (360.3, 854.3) | (15414.5, 24286.4) | (5731.3, 9862.6) | (16381.1, 25528.0) | (6226.6, 10656.6) |
| 90–94 years | Men | 4503.1 | 2277.4 | 1279.7 | 794.3 | 30928.7 | 15556.7 | 32208.5 | 16351.1 |
| | | (3570.8, 5424.8) | (1723.4, 2797.7) | (780.6, 1974.7) | (470.4, 1260.1) | (24525.6, 37259.4) | (11772.3, 19111.2) | (25496.7, 38823.4) | (12430.9, 20137.7) |
| | Women | 3830.3 | 1972.1 | 1201.4 | 736.9 | 26164.8 | 13404.7 | 27366.2 | 14141.6 |
| | | (2850.0, 4695.4) | (1422.3, 2444.6) | (747.8, 7811.3) | (462.1, 7099.3) | (19468.8, 32074.9) | (9667.8, 16616.3) | (20575.8, 33392.8) | (10336.5, 17610.5) |
| ≥95 years | Men | 7142.1 | 4105.7 | 1435.6 | 935.4 | 37795.7 | 21091.7 | 39231.3 | 22027.1 |
| | | (5472.7, 8682.8) | (2986.7, 5131.5) | (797.6, 2333.8) | (512.8, 1608.0) | (28961.5, 45949.0) | (15343.0, 26361.5) | (30067.7, 47736.2) | (16135.2, 27.507.2) |
| | Women | 6521.9 | 4112.0 | 1322.3 | 846.3 | 33913.7 | 21050.3 | 35236.0 | 21896.6 |
| | | (4686.3, 8033.0) | (2902.6, 5201.1) | (781.2, 2101.1) | (494.0, 1341.7) | (24368.6, 41771.8) | (14859.2, 26625.7) | (25435.3, 43325.3) | (15589.1, 27618.1) |

*Abbreviation* UI = uncertainty interval

years and men aged 25–29, 30–34, 35–39, and 40–44 years. During this time, the greatest rate of decline in deaths, YLLs, and DALYs was observed in women and men aged 65–69 years, and women and men aged 80–84 years had the greatest rate of decline in YLDs. Women and men aged ≥95 years exhibited the lowest decline in deaths, YLLs, and DALYs, and women aged 30–34 years and men aged 45–49 years had the lowest decline in YLDs. Details of age- and sex-specific rates of deaths, YLDs, YLLs, and DALYs in 1990 and 2019 are presented in Table 3. The changes in rate between 1990 and 2019 are presented in S5 Table in S1 File.

## Discussion

In this study, we estimated the cardiovascular burden attributable to 13 dietary risk factors in Australia by sex and age groups between 1990 and 2019. In both women and men, ischemic heart disease had the highest burden of specific CVD attributable to dietary risk factors across time. A diet high in red meat for women and a diet low in wholegrains for men were the two leading individual dietary risk factors for diet-related CVD burden in Australia. Our results showed that from 1990 to 2019, the age-standardised rate of diet-related CVD deaths and DALYs decreased significantly in the Australian population with some notable sex and age differences. Between 1990 and 2019, the rate of deaths and DALYs were higher in men than women overall and for each of the dietary risk factors that we investigated. Diet-related CVD burden was greatest in the late-middle aged groups (65–74 years) in both sexes over the years

in whom there was also the greatest rate of decline in the rate for diet-related vascular burden. A diet high in sodium for women and a diet high in processed meat for men exhibited the lowest decrease in CVD burden during this time.

Our findings build on, and further consolidate the evidence that dietary risk factors are among the leading lifestyle determinants for CVD burden in the Australian population, alongside tobacco use, insufficient physical activity, and excessive alcohol consumption [7, 18, 19]. While educational initiatives and programs (e.g. Australian Dietary Guidelines, Health Star Rating) [20] have been implemented by the Australian government over the last few decades, according to the Australian Bureau of Statistics (Health Survey), Australians are still not meeting the recommended intakes of healthy and unhealthy foods [21]. Healthy dietary intake between 1995 and 2011 among Australians improved slightly [22]. This suggests that changes in observed dietary patterns alone may not have been sufficient to have impacted the decrease in diet-related CVD burden as seen in our findings. The significant decreases in CVD burden in the Australian population might be due to additional attributing factors, including but not limited to the decrease in smoking, increased physical activity, and availability of treatment and management for high blood pressure, cholesterol, and other CVD-related morbidities [23]. Our findings demonstrate that high red meat intake was one of the leading dietary risk factors for CVD. However, the evidence that links high red meat intake and CVD is moderate [24].

The burden of CVD-related mortality and morbidity attributable to dietary risk factors was higher in men than women in our study. Previous studies suggest that men have higher rates of cardio-metabolic risk factors, including dyslipidaemia, insulin resistance, and comorbidities such as type 2 diabetes, hypertension, and abdominal obesity [25, 26]; therefore, this may have contributed to the higher rates of CVD-related mortality, morbidity, and overall poor health among men. Diets low in wholegrains, legumes and high in red meat were the major risk factors for CVDs attributable to dietary risk factors for both sexes. This is in line with the evidence suggesting that diets high in wholegrains and legumes offer cardio-protective benefits as a result of their high fibre and micronutrient content [27]. A diet high in red meat, on the other hand, is high in saturated fat, if the consumed meat is not lean, as is the case mostly [28, 29]. High red meat intake, therefore, may increase the risk of cardio-metabolic risk factors and increase inflammatory biomarkers (e.g. C-reactive protein, interleukin-6), which are implicated in the pathophysiology of CVDs [28, 29]. Over the past three decades, the intake of red meat has decreased among the Australian population while that of poultry has dramatically increased [30]. However, reports from 2011/12 show that the average intake of red meat was 565 grams per week per person aged ≥2 years, which exceeds the Heart Foundation recommendation of 350 grams of unprocessed red meat per week [31].

The decline in CVD burden attributable to suboptimal dietary patterns in our study is consistent with the findings from other developed countries [5]. Previous studies have found that low wholegrains intake was the leading diet-related risk factor for non-communicable diseases, including CVDs in high-income countries [5]. Compared to countries or regions with similar socio-demographics or socio-economics (i.e. the United States of America and countries of the European Union), Australia was among the nations with the highest rate of CVD burden for high red meat intake over the last three decades [5]. Our finding is in agreement with previous studies reporting red meat intake being the main contributor for CVD burden, including mortality and DALYs [32]. In contrast, CVD deaths from high sodium intake in Australia remained low between 1990 and 2019 [5]. Similarly, a study in countries of the European region reported that sodium was the fourth leading risk factor for CVD mortality, behind diets low in whole grains, nuts and seeds, and fruits [33]. Whereas a study in Asian countries of similar sociodemographics ranked sodium as the third leading risk factor for CVD burden, again behing a diet low in whole grains, as well as diets low in legumes [34]. In our study, ischemic

heart disease was the leading CVD attributable to dietary risk factors during 1990 and 2019, which is also the leading diet-related CVD globally [35].

To our knowledge, no previous study has investigated the trend of CVD mortality and morbidity attributable to dietary risk factors by sex and age groups in Australia. We used the most robust and recent GBD data to estimate the burden of CVDs attributable to dietary risk factors in the Australian population. However, there were several limitations to this study. Although dietary data utilised for this study came from various sources, there were limited dietary surveys and food balance data available for Australia. Dietary data collected from surveys and observational studies are prone to self-reporting biases and measurement errors. These inherent limitations may have biased analyses and interpretations presented in this study. This indicates that more robust dietary data collection methods are required to continuously report on diet-related CVD in Australia. Of the 15 dietary risk factors included in the GBD, data for 13 dietary risk factors were available for this study (with data for low calcium and milk not available). In the future, it may be warranted to include new and emerging dietary risk factors such as ultra processed foods [36], given their high consumption in Australia and the increasing data on their impact on health. The current GBD data for Australia do not estimate state or territory-specific burden of diseases and risk factors; such data may capture more demographic specific issues informing state and territory governments to undertake more tailored local programmes and policies for healthy nutrition and cardiovascular health. Furthermore, it is important to understand the burden of dietary risk factors in vulnerable groups such as the Aboriginal and Torres Strait Islander people and people representing culturally and linguistically diverse communities. Finally, apart from stratifying for age and gender we have not conducted other sensitivity analyses, future research could include these.

Over the last few decades, numerous national-level lifestyle interventions and policies have aimed to reduce the diet-related CVD burden in Australia [9, 37–39]. The decrease in CVD burden attributable to dietary risk factors over the past three decades may be due to the impact of mass media campaigns, food labelling schemes, food subsidisations, school nutrition policies, and workplace nutrition programs. Previous dietary policies interventions have focused predominantly on limiting saturated fat, sodium and sugar intake, and promoting fruit and vegetable consumption [9]. However, our results suggest that it may be beneficial to promote a diet high in wholegrains and legumes, and low in red meat and trans-fatty acids, in addition to lowering sodium intake. Furthermore, there is a need to develop sex-specific tailored messages and advice around adopting healthier eating patterns, which might be more effective than generic messages. Future studies are warranted to assess the effectiveness of such dietary changes on CVD outcomes, as well as their feasibility and any potential challenges or barriers to their implementation.

## Conclusions

Although the burden of CVDs attributable to dietary risk factors decreased over the past 30 years, a diet low in wholegrains and high in red meat remain a concern for both men and women in Australia. Ischemic heart disease was the most prominent diet-related CVD. Our study highlights the need for dietary policies to focus on a whole of diet approach emphasising wholegrains and legumes and reducing red meat and with additional efforts to target those at higher risk including men and older adults.

## Supporting information

**S1 File.**
(DOCX)

## Acknowledgments

**Institute for Physical Activity and Nutrition** (IPAN), Heart Foundation, National Health & Medical Research Council (NHMRC)

## Author Contributions

**Writing – original draft:** Sebastian V. Moreno.

**Writing – review & editing:** Sebastian V. Moreno, Riaz Uddin, Sarah A. McNaughton, Katherine M. Livingstone, Elena S. George, George Siopis, Ralph Maddison, Rachel R. Huxley, Sheikh Mohammed Shariful Islam.

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
