## [Decision Letter · Decision Letter 0]

31 Aug 2023

PONE-D-23-01663The burden of cardiovascular disease attributable to dietary risk factors in Australia between 1990 and 2019PLOS ONE

Dear Dr. Siopis,

Thank you for submitting your manuscript to PLOS ONE. After careful consideration, we feel that it has merit but does not fully meet PLOS ONE’s publication criteria as it currently stands. Therefore, we invite you to submit a revised version of the manuscript that addresses the points raised during the review process.

We look forward to receiving your revised manuscript.

Kind regards,

Amir Hossein Behnoush

Academic Editor

PLOS ONE

“The GBD Study is funded by the Bill and Melinda Gates Foundation.”

“The authors received no specific funding for this work.”

4. We notice that your supplementary tables are included in the manuscript file. Please remove them and upload them with the file type 'Supporting Information'. Please ensure that each Supporting Information file has a legend listed in the manuscript after the references list.

Additional Editor Comments:

The authors are encouraged to provide MAJOR REVISIONS, based on reviewers' comments. The points mentioned by both reviewers (especially reviewer #2) should be responded point-by-point.

Reviewers' comments:

Reviewer's Responses to Questions

**Comments to the Author**

1. Is the manuscript technically sound, and do the data support the conclusions?

Reviewer #1: Yes

Reviewer #2: Yes

2. Has the statistical analysis been performed appropriately and rigorously? 

Reviewer #1: I Don't Know

Reviewer #2: Yes

3. Have the authors made all data underlying the findings in their manuscript fully available?

Reviewer #1: Yes

Reviewer #2: Yes

4. Is the manuscript presented in an intelligible fashion and written in standard English?

Reviewer #1: Yes

Reviewer #2: Yes

5. Review Comments to the Author

Reviewer #1: This interesting paper uses global burden of disease methodology to assess the contribution of poor diet to CVD risk in Australia over the period 1990-2019.

Note that it would be helpful in reviewing this work if line numbers were included.

The results are valuable to inform health promotion messages.

The analysis is explained in detail, but I am not able to adequately assess the detail of the methodology.

“Of the 13 dietary risk factors assessed, diets high in sodium for women and diets high in processed meat for men observed the lowest decrease in percentage change for age-standardised rate for CVD deaths,” This is from page 6 and I wonder whether there is something wrong here with a decrease in percentage change.

In the second para of the discussion changes in diet in Aus might not be sufficient to account for changes in diet-related CVD burden. Does this make sense in terms of the way the GBD works?

Figure 5 in the Whitnall and Pitts reference shows that intake of sheep and beef meat declined in Aus and poultry intake increased dramatically during 1998-2018 which does not seem consistent with the statement “Over the past three decades, the intake of red meat has increased among the Australian population, 30” attributed to this reference.

Bottom of page 9-page 10, here you talk about targeting “diet low in sodium, wholegrains, legumes, and high in red meat and trans-fatty acids,” why is sodium grouped with things to eat more of rather than things to eat less of?

Reviewer #2: Overall, this manuscript presents an important investigation into the burden of cardiovascular diseases (CVDs) attributed to dietary risk factors in Australia over a 30-year period. While the study is valuable and well-structured, there are several areas where improvements are needed before it can be considered for publication. The review is constructive and provides specific recommendations for enhancing the manuscript.

1. Clarity and Organization:

- The manuscript's overall structure is clear, with sections such as the abstract and introduction providing a concise overview. However, the presentation of results and discussion can be improved for better clarity and coherence. It would be beneficial to reorganize and rewrite certain sections to ensure a smoother flow of information.

2. Data Sources and Limitations:

- The manuscript mentions that dietary data were sourced from various surveys and organizations. However, the authors should provide more specific details regarding these sources, including their representativeness, sample sizes, and potential biases. Understanding the quality of the dietary data is critical for the reader's confidence in the results.

- The limitations section should be expanded to discuss how the limitations in dietary data may have influenced the findings and the interpretation of the study. Addressing these limitations would enhance the manuscript's transparency.

3. Statistical Methods and Uncertainty:

- While the manuscript explains the statistical methods employed, it would be beneficial to elaborate on the assumptions made during modeling and discuss the potential implications of these assumptions on the results. Providing a more comprehensive discussion of uncertainties would enhance the robustness of the findings.

4. Comparison with Other Studies:

- The manuscript claims to be the first to investigate trends in CVDs attributable to dietary risk factors in Australia. While this is an important contribution, it would be valuable to contextualize the findings by comparing them with similar studies in other countries or regions. Discussing the consistency or divergence of results would provide a broader perspective.

5. Sex and Age Differences:

- The manuscript touches upon differences in CVD burden between sexes and age groups but does not delve deeply into the potential reasons for these disparities. It would be beneficial to include discussions about the socio-cultural, economic, and behavioral factors that might contribute to these differences, providing a more comprehensive analysis.

6. Policy Implications:

- The manuscript concludes by suggesting the need for dietary policies but lacks specific recommendations or discussions on the feasibility and potential challenges of implementing such policies. A more detailed exploration of the implications of the findings for public health policy and potential interventions is warranted.

7. Accessibility and Reproducibility:

- The manuscript should include a clear reference to the data sources and analytical methods used to ensure transparency and reproducibility. This is crucial for researchers who may wish to replicate or build upon the study.

Overall, this manuscript makes a valuable contribution to understanding the burden of CVDs attributable to dietary risk factors in Australia. However, to enhance its quality and readiness for publication, significant revisions are necessary. The authors should consider reorganizing and rewriting certain sections, providing more specific details about data sources and limitations, conducting sensitivity analyses, comparing findings with other studies, exploring reasons for sex and age differences, and discussing policy implications in greater depth. Addressing these recommendations will strengthen the manuscript and its potential impact on public health.

6. PLOS authors have the option to publish the peer review history of their article (what does this mean?). If published, this will include your full peer review and any attached files.

Reviewer #1: No

Reviewer #2: No

---

## [Author Response · Author response to Decision Letter 0]

2 Nov 2023

Please see "Responses to Reviewers" attached.

---

## [Decision Letter · Decision Letter 1]

20 Nov 2023

The burden of cardiovascular disease attributable to dietary risk factors in Australia between 1990 and 2019

PONE-D-23-01663R1

Dear Dr. Siopis,

We’re pleased to inform you that your manuscript has been judged scientifically suitable for publication and will be formally accepted for publication once it meets all outstanding technical requirements.

Kind regards,

Amir Hossein Behnoush

Academic Editor

PLOS ONE

Additional Editor Comments (optional):

Reviewers' comments:

Reviewer's Responses to Questions

**Comments to the Author**

1. If the authors have adequately addressed your comments raised in a previous round of review and you feel that this manuscript is now acceptable for publication, you may indicate that here to bypass the “Comments to the Author” section, enter your conflict of interest statement in the “Confidential to Editor” section, and submit your "Accept" recommendation.

Reviewer #1: All comments have been addressed

Reviewer #2: All comments have been addressed

2. Is the manuscript technically sound, and do the data support the conclusions?

Reviewer #1: (No Response)

Reviewer #2: Yes

3. Has the statistical analysis been performed appropriately and rigorously? 

Reviewer #1: (No Response)

Reviewer #2: Yes

4. Have the authors made all data underlying the findings in their manuscript fully available?

Reviewer #1: (No Response)

Reviewer #2: No

5. Is the manuscript presented in an intelligible fashion and written in standard English?

Reviewer #1: (No Response)

Reviewer #2: Yes

6. Review Comments to the Author

Reviewer #1: (No Response)

Reviewer #2: (No Response)

7. PLOS authors have the option to publish the peer review history of their article (what does this mean?). If published, this will include your full peer review and any attached files.

Reviewer #1: No

Reviewer #2: **Yes: **Mohsen Abbasi-Kangevari

---

## [Editor Report · Acceptance letter]

28 Nov 2023

PONE-D-23-01663R1 

The burden of cardiovascular disease attributable to dietary risk factors in Australia between 1990 and 2019 

Dear Dr. Siopis:

I'm pleased to inform you that your manuscript has been deemed suitable for publication in PLOS ONE. Congratulations! Your manuscript is now with our production department. 

Kind regards, 

on behalf of

Dr. Amir Hossein Behnoush 

Academic Editor

PLOS ONE